# Numerical and Experimental Studies for Fatigue Damage Accumulation of CFRP Cross-Ply Laminates Based on Entropy Failure Criterion

**DOI:** 10.3390/ma16010388

**Published:** 2022-12-31

**Authors:** Huachao Deng, Asa Mochizuki, Mohammad Fikry, Shun Abe, Shinji Ogihara, Jun Koyanagi

**Affiliations:** 1Department of Materials Science and Technology, Tokyo University of Science, Tokyo 125-8585, Japan; 2Department of Mechanical Engineering, Tokyo University of Science, Chiba 278-8510, Japan

**Keywords:** numerical simulation, composite laminates, entropy-based strength degradation, CFRP transverse cracking behavior, fatigue

## Abstract

The transverse cracking behavior of a carbon-fiber-reinforced plastic (CFRP) cross-ply laminate is investigated using a fatigue test and an entropy-based failure criterion in this study. The results of fatigue experiments show that the crack accumulation behavior depends on the cyclic number level and frequency, in which two obvious transverse cracks are observed after 10^4^ cyclic loads and 37 transverse cracks occur after 10^5^ cycles. The final numbers of transverse cracks decrease from 29 to 11 when the load frequency increases from 5 Hz to 10 Hz. An entropy-based failure criterion is proposed to predict the long-term lifetime of laminates under cyclic loadings. The transverse strength of 90° ply is approximated by the Weibull distribution for a realistic simulation. Progressive damage and transverse cracking behavior in CFRP ply can be reproduced due to entropy generation and strength degradation. The effects of stress level and load frequency on the transverse cracking behavior are investigated. It is discovered that, at the edge, the stress σ_22_ + σ_33_ that is a dominant factor for matrix tensile failure mode is greater than the interior at the first cycle load, and as stress levels rise, a transverse initial crack forms sooner. However, the initial transverse crack initiation is delayed as load frequencies increase. In addition, transverse crack density increases quickly after initial crack formation and then increases slowly with the number of load cycles. The proposed method’s results agree well with those of the existing experimental method qualitatively. In addition, the proposed entropy-based failure criterion can account for the effect of load frequency on transverse crack growth rate, which cannot be addressed by the well-known Paris law.

## 1. Introduction

With developments in recent years, composite materials made from two or more constituent materials, such as aramid-fiber-reinforced plastic (FRP) [1,2], glass FRP [3], basalt-fiber-reinforced polymer [4] and carbon-fiber-reinforced plastics (CFRP) [5,6], have been extensively applied in automobile, aerospace, civilian and military industries due to its excellent corrosion resistance, fatigue resistance and creep resistance [7,8]. By replacing traditional advanced aluminum alloys, a weight saving of up to one and a half tons can be obtained in the A380 [9], whose central wing box is made of CFRPs. In practical applications, the CFRPs usually suffer from post-buckling under eccentric force [10,11,12] and cyclic loadings. Nevertheless, the failure mechanism of CFRP typically consists of micro failures in the fiber, matrix and interface between the fiber and matrix [13,14,15]. Among these, matrix failure usually exhibits time dependency [16,17,18]. Thus, accurate predictions of the long-term lifetime of CFRP under cyclic loadings are of great significance.

Initially, the lifetime of CFRPs was typically investigated using an experimental method [19,20,21,22]. Reifsnider et al. [23] found that the fatigue failure of fiber-reinforced plastics usually consisted of transverse crack multiplication, delamination propagation and fiber breakage. It is worth noting that the transverse crack multiplication and delamination propagation account for a large portion of the total period until fatigue life due to the fact that the final failure of CFRP is caused by fiber breakage. Therefore, some investigations into transverse cracking behavior of CFRP were reported. Ogihara et al. [24] studied the influence of stacking sequence on quasi-isotropic CFRP laminates’ microscopic fatigue damage mechanism, and it was observed that transverse crack density depends on the stress level and number of loading cycles. Yokozeki et al. [25] conducted the investigations into transverse crack propagations, in which transverse crack initiation and propagation in the width direction of cross-ply laminates were observed. Hosoi et al. [26] quantitatively evaluated the effect of the applied load level on fatigue damage propagation phenomenon, and a modified version of Paris’ law was proposed to determine damage propagation. Kitagawa et al. [27] recently proposed a tension-tension fatigue test method to investigate the transverse cracking behavior of CFRP ply. The applied stress level had a significant impact on damage accumulation [27], and oblique cracks usually started near transverse cracks. Li et al. [28] investigated the effect of various factors, such as elevated temperature and hydraulic pressure, on the property evolution for a carbon/glass hybrid rod. In addition, the frequency of load on composite materials is also an important factor to predict long-term lifetime [29,30].

Experimental investigations are often expensive and time consuming. In addition, results can be affected by the size of the specimen, temperature, humidity and loading conditions. Therefore, researchers are trying to search for alternative ways to predict fatigue behavior efficiently. In addition to the well-known stress-based or energy-based methods for the estimation of fatigue life [31,32,33,34], approaches based on irreversible thermodynamics [35,36,37] were also proposed to investigate the failure mechanism and long-term lifetime of solid materials. It is well known that not only irreversible microplastic deformation but also internal friction can result in permanent degradations, such as in plastic. In the view of thermodynamics, these irreversible degradations can be measured by entropy, which is a non-negative quantity and can serve as a basis for the damage evolution metric for elastic and inelastic deformations. When the entropy generation of a material reaches a threshold value called fracture fatigue entropy (FFE) [38,39], final failure occurs. Many publications have been reported to show that the estimation of fatigue life based on entropy is promising. It should be noted that the fracture fatigue entropy of material is also constant, even in the case where A656-grade steel is subjected to ultrasonic vibration at 20 kHz [40].

Although the entropy-based failure criterion has been widely utilized to successfully estimate the fatigue life of metal components, its application to investigate the long-term lifetime of CFRP under cyclic loading is limited. Huang et al. [38] investigated the effect of stacking sequences on the internal friction and fracture fatigue entropy of CFRP ply. Moreover, with the consideration of both confidence level and reliability, the fatigue life estimation of CFRP was determined [39]. Koyanagi et al. [41,42,43,44] recently developed a computational framework with entropy damage to study the failure process of a viscoelastic matrix. However, no research has been published on the transverse cracking behavior of CFRP laminate, which is a dominant failure mode.

In this study, both experimental and numerical methods are utilized to investigate the transverse cracking behavior of CFRP laminates under cyclic load. The fatigue experiment results reveal that cyclic load level and frequency are two key factors that affect crack accumulation behavior. To study the transverse cracking behavior of CFRP laminate subjected to cyclic loadings, a criterion based on entropy is proposed. The model of this study presents strength and fracture energy reduction based on the stress–strain history. In addition, Hashin’s failure criterion [8] is adopted to consider fiber and matrix compressive damage initiation criteria. The entropy-based failure criterion takes into account the effect of load frequency on transverse cracking behavior that is not considered by Paris’ law. The proposed method will be extended to simulate transverse initiation and propagation under random loadings.

## 2. Experimental Study

### 2.1. Material Manufacturing and Damage Observation

The material used in this study is carbon fiber/epoxy unidirectional (UD) tape prepreg (Torayca, T700SC/2592, 0.14 mm/ply). The prepregs are cured in an autoclave at a temperature of 130 °C and pressure of 0.2 MPa. The stacking configuration is [0°/90°_3_]_s_; the post cured thickness of the laminate is about 1.15 mm. The specimen’s measurement is as shown in Figure 1. Laminates are cut into the measurement by using a composite material cutting machine (AC-300CF, Maruto Testing Machine). GFRP tabs are glued to the specimen ends by using adhesive glue before the testing. As CFRP laminates are opaque, crack observation is done using X-ray radiography. An X-ray machine, M-100S, SOFTEX is used with the applied voltage and current of 14 KVP and 1.5 mA, respectively (exposure time is 3 min). This damage observation method is a common method used to detect transverse cracks and delamination in the CFRP laminates [45,46,47,48].

### 2.2. Fatigue Loading

To determine the stress ratio used in fatigue testing of this study, six specimens from the same manufacturing batch used for fatigue test are monotonically and cyclically loaded. Both of the experiments are performed with the cross-head displacement speed of 1 mm/min using Tensilon RTF-1350 A & D tensile test machine (Shimadzu, Kyoto, Japan). From the testing, we obtained the laminate’s average maximum tensile strength of 647 MPa where the transverse cracks initiated from about the stress level of 250 MPa.

The fatigue test of CFRP is performed at room temperature, and the relationships between transverse cracks and load conditions, such as cyclic load number and frequency, are investigated. The fatigue loading machine (Shimadzu, Kyoto, Japan, EHF-LV020K1A) used in this study is shown in Figure 2. The tension-tension sinusoidal fatigue load used in the tests is shown in Figure 3, in which the stress ratio is R (R = σ*_min_*/σ*_max_*) fixed at 0.1, where σ*_min_* = 20 MPa and σ*_max_* = 200 MPa (30% of the tensile strength). The transverse crack density in this study is computed by dividing the number of transverse cracks by the certain area of length 60 mm in the specimen’s center as shown in Figure 4.

Figure 5 shows the typical distribution of transverse cracks under various cyclic loads with a fixed load frequency of 1 Hz. It is found that there is no transverse crack when the cyclic load number is less than 10^3^. As the cyclic load number increases to 10^4^, two obvious transverse cracks are observed. The fatigue test is terminated when the cyclic load number increases to 10^5^ and the number of transverse cracks is 37. Furthermore, Figure 6 depicts the transverse crack behaviors obtained after 10^5^ cyclic loads at various load frequencies. In contrast to the effect of the cyclic load number, as the load frequency increases from 5 Hz to 10 Hz, the final numbers of transverse cracks decrease from 29 to 11. Based on the above fatigue experimental results, it is concluded that cyclic load number and frequency are two significant factors that affect the transverse cracks of CFRP. These will be further investigated by a new numerical method based on an entropy-based failure criterion in the next section.

## 3. Numerical Study

### 3.1. Orthotropic Viscoelastic Model

The analytical relaxation modulus of matrix resin can be found in references [16,41,49], but it is usually characterized by the Maxwell model for numerical simulation. Koyanagi et al. [41] investigated the influence of Maxwell element number on the relaxation modulus and found that sufficient accuracy can be ensured when the Maxwell element number is increased to five. Thus, in this study, five Maxwell elements aligned in parallel shown in Figure 7 are adopted to model viscoelastic matrix resin, in which each element consists of the spring $C_{d}^{k}$ and dashpot $\eta^{k}$ (*k* = 1~5). Stress relaxation is modeled individually in each line, and stress is computed as the sum of all elastic stresses $\sigma^{k}$. Owing to the orthotropic properties of CFRP, 5 × 6 elasticity constants and 5 × 6 viscosity constants are introduced to model six independent stress–strain relationships, and the Hashin’s failure criteria are utilized to determine the damage onset. The material parameters are adopted from the reference [41] and listed in Table 1 and Table 2 since the material properties of the specimen used for the fatigue test have not been determined.

### 3.2. Implementation of Entropy-Based Failure Criterion

The orthotropic viscoelastic model considering the entropy-based failure criterion is implemented into the finite element analysis software Abaqus 2020 by the user subroutine UMAT [41]. At the beginning of time increment *m*, total strain increment $\Delta \varepsilon_{t}^{k,m,0}$ and historical variables, such as elastic strain $\varepsilon_{e}^{k,m-1}$ and damage parameters $d_{f}^{k,m-1}$, are passed into UMAT, and stress at integration point **σ** will be updated by the user-defined constitutive law. In this study, trial stress at the *n*-th (*n* ≥ 1) iteration of increment *k* is computed as
(1)$$\sigma^{k,m,n}=C_{d}^{k}\varepsilon_{e}^{k,m,n}=C_{d}^{k}\left(\varepsilon_{e}^{k,m,n-1}+\Delta \varepsilon_{e}^{k,m,n}\right)$$

In Equation (1), the superscripts *k*, *m*, *n* of σ or ε represent the numbers of Maxwell model, increment and iteration, respectively. The damaged stiffness tensor of *k*-th spring $C_{d}^{k}$ is defined as
(2)$$C_{d,n}^{k}=\left[\begin{array}{cccccc} C_{d,11}^{k} & C_{d,12}^{k} & C_{d,13}^{k} & 0 & 0 & 0\\ & C_{d,22}^{k} & C_{d,23}^{k} & 0 & 0 & 0\\ & & C_{d,33}^{k} & 0 & 0 & 0\\ & & & C_{d,44}^{k} & 0 & 0\\ & \mathrm{Symmetrical} & & & C_{d,55}^{k} & 0\\ & & & & & C_{d,66}^{k} \end{array}\right]$$
where $C_{d,11}^{k}=\left(1-d_{f,n}\right)C_{11}^{k}$, $C_{d,12}^{k}=\left(1-d_{f,n}\right)\left(1-d_{m,n}\right)C_{12}^{k}$, $C_{d,13}^{k}=\left(1-d_{f,n}\right)\left(1-d_{m,n}\right)C_{13}^{k}$, $C_{d,22}^{k}=\left(1-d_{m,n}\right)C_{22}^{k}$, $C_{d,23}^{k}=\left(1-d_{m,n}\right)C_{23}^{k}$, $C_{d,33}^{k}=\left(1-d_{m,n}\right)C_{33}^{k}$, $C_{d,44}^{k}=\left(1-d_{f,n}\right)\left(1-d_{m,n}\right)C_{44}^{k}$, $C_{d,55}^{k}=\left(1-d_{f,n}\right)\left(1-d_{m,n}\right)C_{55}^{k}$, $C_{d,66}^{k}=\left(1-d_{m,n}\right)C_{66}^{k}$, $d_{f,n}$ and $d_{m,n}$ are fiber and matrix damage variables, $C_{ij}^{k}$ is the components of the stress–strain law in intact material and listed in Table 1.

The elastic strain increment $\Delta \varepsilon_{e}^{k,m,n}$ is determined as
(3)$$\Delta \varepsilon_{e}^{k,m,n}=\Delta \varepsilon_{t}^{k,m,n}-\Delta \varepsilon_{v}^{k,m,n}$$
where the viscoelastic strain increment $\Delta \varepsilon_{v}^{k,m,n}$ is expressed as
(4)$$\Delta \varepsilon_{v}^{k,m,n}=\frac{C_{d}^{k}\sigma^{k,m,n}}{\eta^{k}}\Delta t$$

Due to the nonlinear nature of viscoelastic behavior, stress $\sigma^{k,m,n}$ is usually determined by iteration procedure [41] listed in Figure 8.

The increment of dissipated energy $\Delta W_{n}$ and total dissipated energy are determined as follows:(5)$$\Delta W_{n}=\sum_{n=1}^{5}C_{d,n}^{k}\varepsilon_{e}^{k,m,n}\cdot \Delta \varepsilon_{v}^{k,m,n}$$
(6)$$W_{n}=W_{n-1}+\Delta W_{n}$$

After determining the dissipated energy, the entropy generation *s* is computed as
(7)$$s=\frac{W}{T}$$
where *T* is the temperature. Entropy generation *s* is applied to depict the degradation of strength and fracture energy of CFRPs. The strength degradation parameters **α** (*α*_AT_, *α*_AT_ and *α*_O_) [8,41] are taken as a slightly larger constant 300,000 K·mm^3^/J, which will be further studied in next plan. Figure 9 demonstrates the damage onset and evolution of the fiber tensile failure mode [41]. It is also suitable for the other three failure modes, i.e., fiber compressive mode (*e*_fc_), transverse directional tensile mode (*e*_mt_) and transverse compressive mode (*e*_mc_), by computing the corresponding equivalent failure displacement and stress [41].

### 3.3. Results and Discussions

The proposed method is utilized to simulate the transverse cracking behavior of CFRP structures. As shown in Figure 10a, the dimension of the CFRP structure is 100 mm × 10 mm × 6 mm and the symmetrical boundary conditions are applied on surfaces ABCD, CDHG and AEHD. In the 90° layer, the transverse tensile strength is assumed to satisfy the cumulative distribution function for the Weibull distribution $\sigma =\sigma_{0}{\left(\ln\frac{1}{R-1}\right)}^{1/m}$, where *m* = 20 and σ_0_ = 90 MPa. A finite element model made of 7777 nodes and 6000 C3D8 elements is shown in Figure 10b, in which color denotes material property and the size of element is 1 mm × 1 mm × 1 mm. In this example, the transverse crack density is determined by dividing the number of transverse cracks by the length (100 mm). To ensure numerical stability, the stress boundary condition is replaced by applying strain boundary conditions, i.e., the displacement boundary condition is employed on surface ABFE in Figure 10a. The effect of stress levels on results can be reflected by changing the displacement conditions.

The effect of the cyclic load number on the transverse cracking behavior is investigated first. Figure 11 shows the distributions of damage onset index and stress σ_22_ + σ_33_ in the 90° layer after certain cyclic loadings when the applied strain is 0.6%, i.e., U*x* = 0.6 mm, and the load frequency is fixed at 5 Hz. It is found that damage onset index is smaller than one and stress σ_22_ + σ_33_ at the edge is greater than interior at the first cycle load. Note that σ_22_ + σ_33_ is the dominant factor for matrix tensile failure mode [39], so we focus on this value in this study. After 63 cyclic loads, the damage onset index reaches to 1 on the edge, and the stress σ_22_ + σ_33_ in damaged area decreases to a lower value. As the cyclic load increases to 71, 73, the number of cracks is 6, 9, respectively. The phenomenon that the number of transverse cracks increases with cyclic load can also be found in current experimental results. For a further investigation, the evolutions of the damage onset index, damage evolution index and stress σ_22_ + σ_33_ at the first initiated cracked element versus the cyclic load number are shown in Figure 12. It is found that the damage onset index increases after the first load cycle and gradually reaches 1 after 63 cyclic loads. After damage occurs, the damage evolution index becomes 1 instantly without any evolution due to the higher stress level, and stress σ_22_ + σ_33_ drops down to a lower value. This phenomenon can be also found in the previous works [41] of the authors in this study. It is worth noting that owing to viscoelastic behavior of resin matrix, the maximum stress σ_22_ + σ_33_ gradually decreases with cyclic number.

Figure 13 shows the effect of stress level on the transverse cracking behavior, in which the applied strain is increased from 0.7% to 0.8%, i.e., displacement boundary conditions applied on the face ABFE are 0.7 mm and 0.8 mm, respectively. It should be noted that the initial transverse cracks form after 38 cyclic loads when the strain boundary condition is 0.7% and after 23 when it is 0.8%. This can be explained by the fact that a higher stress level will lead to earlier failure. In addition, the transverse crack density increases quickly after initial crack formation and then increases slowly. This phenomenon can also be found in the experimental results of Kitagawa et al. [27], and the crack density versus stress level obtained by the proposed method agrees qualitatively with those in reference [27].

In addition to stress level, the load frequency is also investigated by the proposed method finally. Figure 14 shows the variation of transverse crack density versus the number of load cycles under different load frequencies, where the given strain condition of the whole specimen is 0.8%. Differing from the effect of stress level, as load frequency increases, initial transverse crack initiation is delayed. Initial cracks are formed after 13 cyclic loads when the frequency is 2.5 Hz; 5 Hz and 10 Hz have 23 and 44 cyclic loads, respectively. It is worth noting that, although some empirical formulations, such as Paris’ law [27] based on the energy release rate, are proposed to predict the crack accumulation phenomenon, the effect of load frequency on transverse cracking behavior cannot be addressed. However, the effect of load frequency can be addressed by the proposed entropy-based failure criterion. Thus, the proposed method can account for the effect of load frequency on transverse crack growth behaviors of CFRPs better than the conventional methods.

### 3.4. Summary and Future Plan

Based on the above discussions, it is believed that the proposed failure model can characterize the dependency of cyclic load number, stress level and load frequency on transverse cracking behavior. Only 100 cyclic loads are considered owing to the high computational cost of three-dimensional simulation. It is necessary to develop a multi-timescale computational framework to reduce the time cost [50] in the next plan. Owing to the viscoelastic nature of polymeric matrix, the heat generation phenomenon under cyclic load is a key factor in estimating the fatigue life of CFRPs [51,52,53,54,55], but it is still not implemented into the proposed entropy-based failure criterion. Although the proposed method is only applied to the transverse cracking behavior of cross-ply laminates in this study, the quasi-isotropic laminates [56,57,58,59,60], delamination caused by the transverse cracks and components in the civil engineering [61] can also be analyzed. In addition, stiffness degradation caused by entropy generation should also be considered [62].

## 4. Conclusions

The transverse cracking behavior of CFRP ply under cyclic load is investigated using experimental and numerical methods. It is concluded that:(1)Based on experimental results, it is found that cyclic load level and frequency are two key factors that affect the crack accumulation behavior. Two obvious transverse cracks are observed after 10^4^ cyclic loads and 37 transverse cracks occur after 10^5^ cycles in the experimental test. The final numbers of transverse cracks decrease from 29 to 11 when the load frequency increases from 5 Hz to 10 Hz.(2)To predict the long-term lifetime of CFRP laminate under fatigue loads, an entropy-based failure criterion is proposed. Progressive damage and transverse cracking behavior in CFRP ply are simulated. Numerical results show that as stress levels rise, transverse initial cracks form earlier, whereas initial transverse crack formation slows as load frequency rises. When the load frequency is fixed as 5 Hz, the initial transverse cracks form after 63 cyclic loads when the strain boundary condition is 0.6%, and those of 0.7% and 0.8% are 38 and 23.(3)In addition, as load frequency increases from 2.5 Hz to 10 Hz, the numbers of cyclic loads where the initial crack forms increase from 13 to 44. Comparing the proposed failure model to reference results demonstrates that it can account for the effects of cyclic load number, stress level and load frequency on transverse cracking behavior.(4)The proposed entropy-based failure criterion can model the effect of load frequency on transverse cracking behavior that cannot be addressed by Paris’ law. This may be a significant contribution to this study. However, many studies, such as those on more efficient computational frameworks and heat generation under cyclic loading, should be conducted in order to accurately predict the lifetime of CFRPs under cyclic loading. These will be presented in future studies.

## Figures and Tables

**Figure 1 materials-16-00388-f001:**
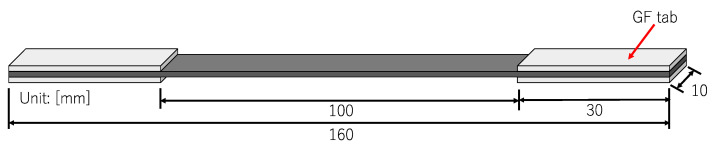
Specimen used for fatigue test.

**Figure 2 materials-16-00388-f002:**
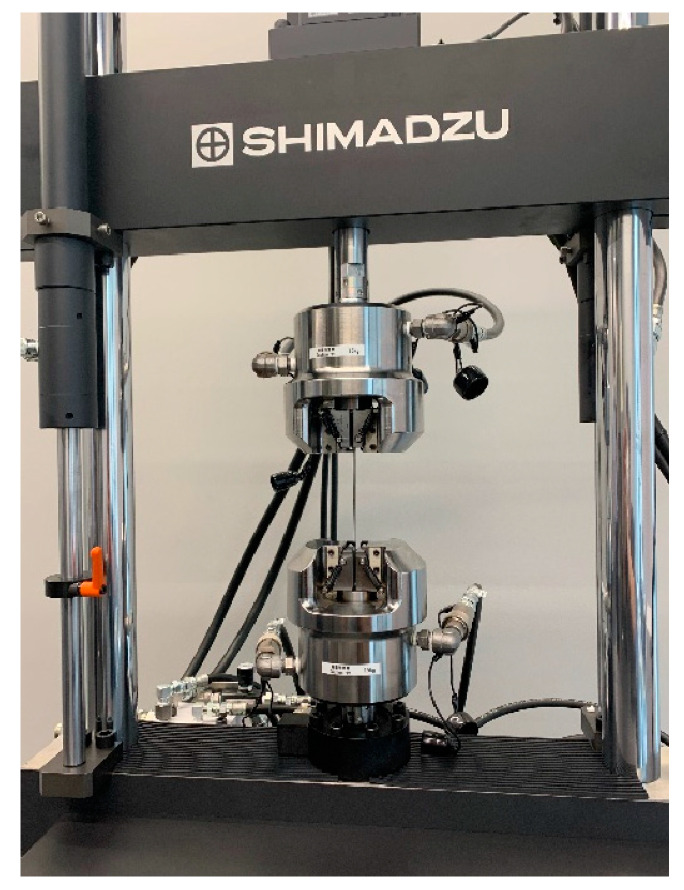
Fatigue loading machine (Shimadzu, EHF-LV020K1A).

**Figure 3 materials-16-00388-f003:**
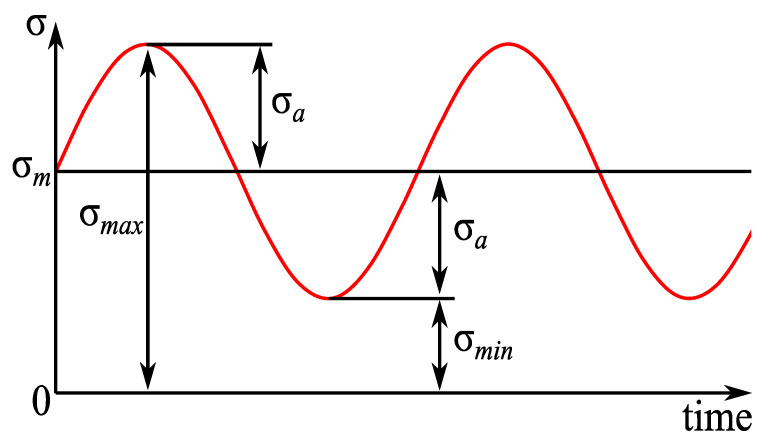
Fatigue cyclic load.

**Figure 4 materials-16-00388-f004:**
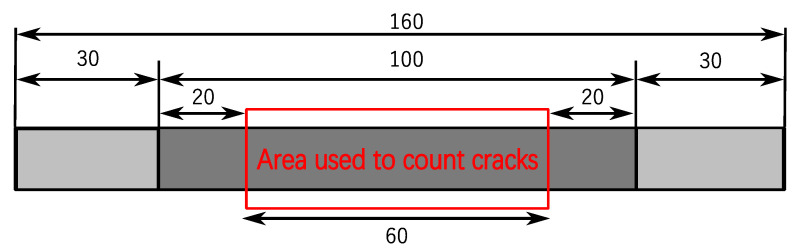
Area to count the number of transverse cracks (Unit: mm).

**Figure 5 materials-16-00388-f005:**
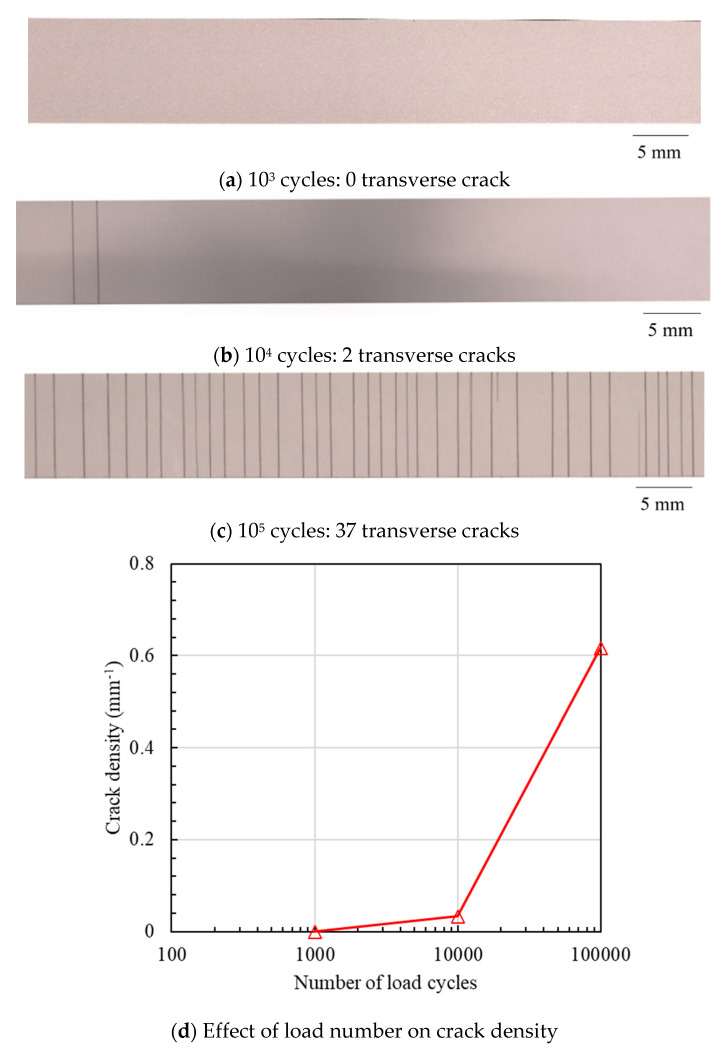
Transverse crack distribution under various cyclic load numbers at a maximum stress of 30% of the tensile strength.

**Figure 6 materials-16-00388-f006:**
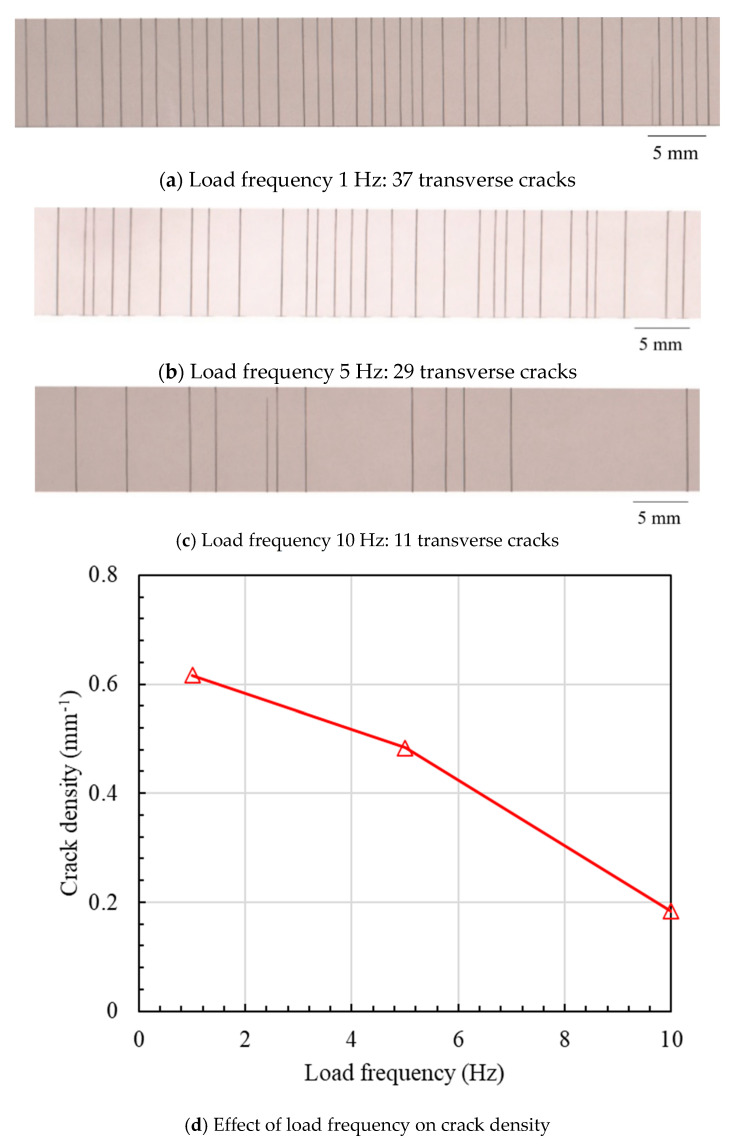
Transverse crack distribution under various load frequencies (10^5^ cycles).

**Figure 7 materials-16-00388-f007:**
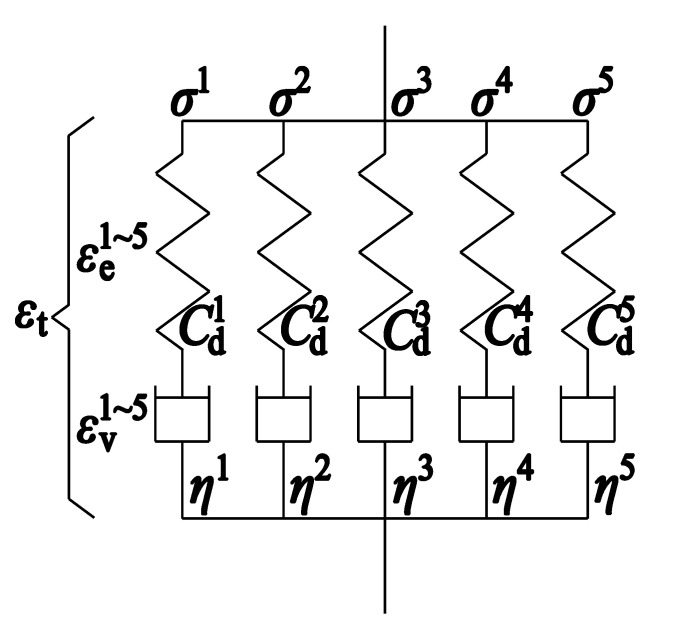
Five Maxwell elements aligned in parallel are adopted to model viscoelastic resin.

**Figure 8 materials-16-00388-f008:**
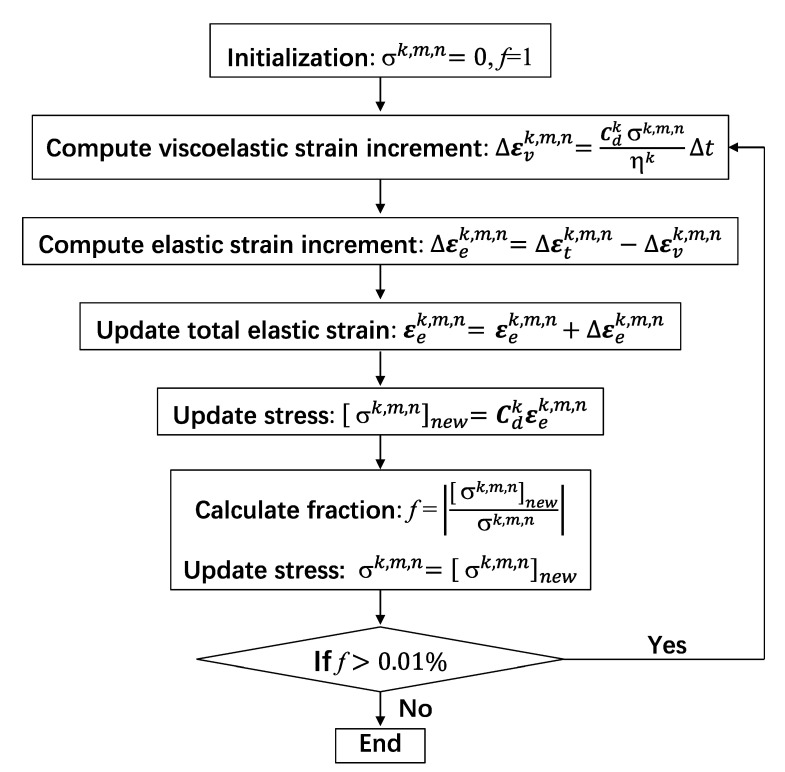
Flowchart for updating stress at the integration point.

**Figure 9 materials-16-00388-f009:**
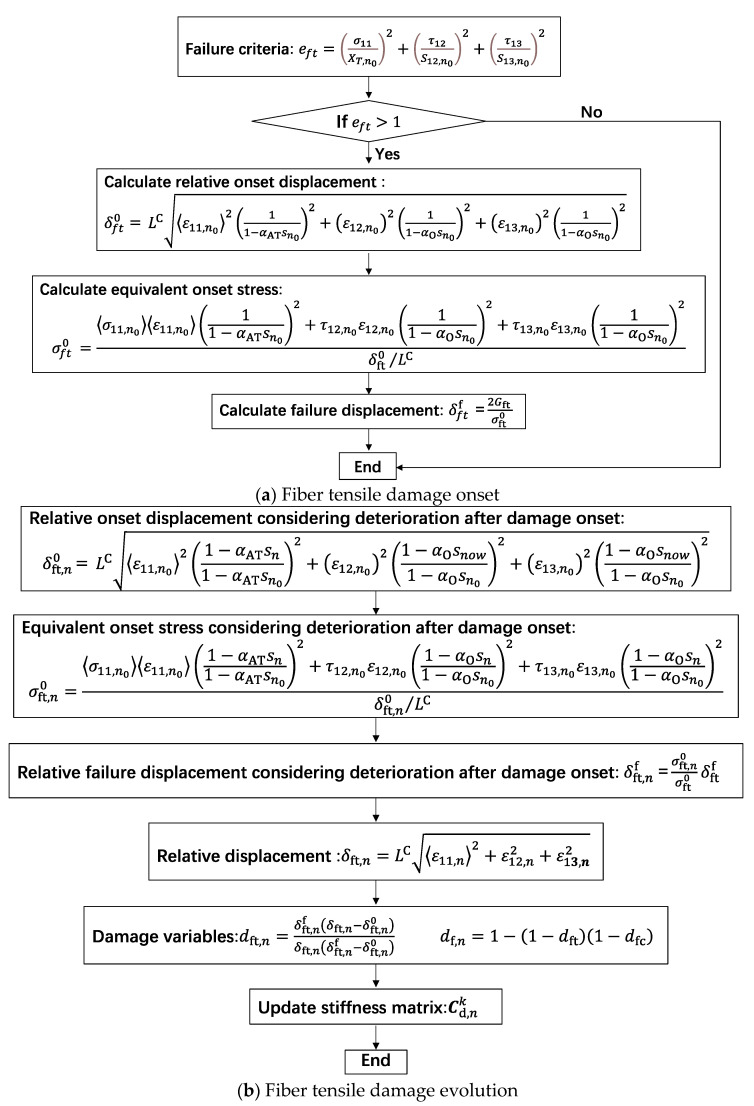
Flowchart for damage onset and evolution of fiber tensile failure mode.

**Figure 10 materials-16-00388-f010:**
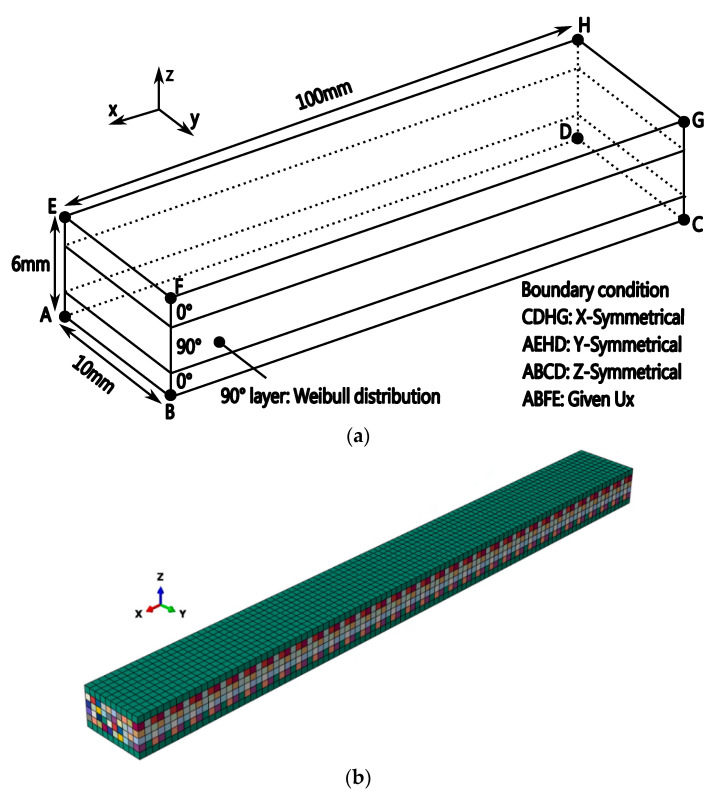
Cross-ply CFRP structures subjected to cyclic loading. (**a**) Geometric model under cyclic loading: symmetric boundary conditions are applied on face ABCD, CDHG and AEHD, respectively; (**b**) Finite element model (color denotes the material property).

**Figure 11 materials-16-00388-f011:**
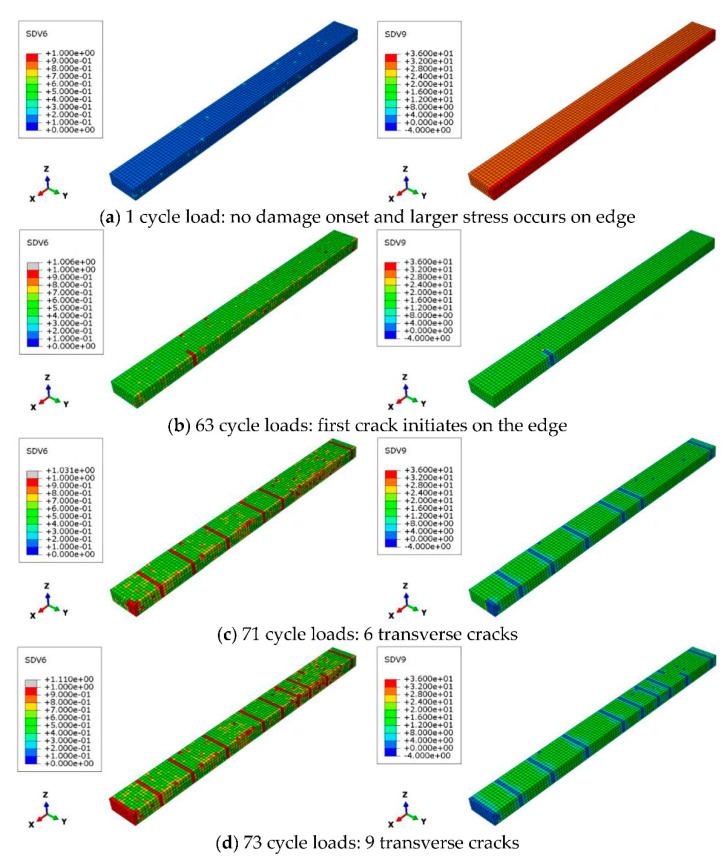
Matrix damage evolution of the 90° layer when the strain boundary condition is 0.6%: left column is the damage onset index, right column is stress σ_22_+σ_33_.

**Figure 12 materials-16-00388-f012:**
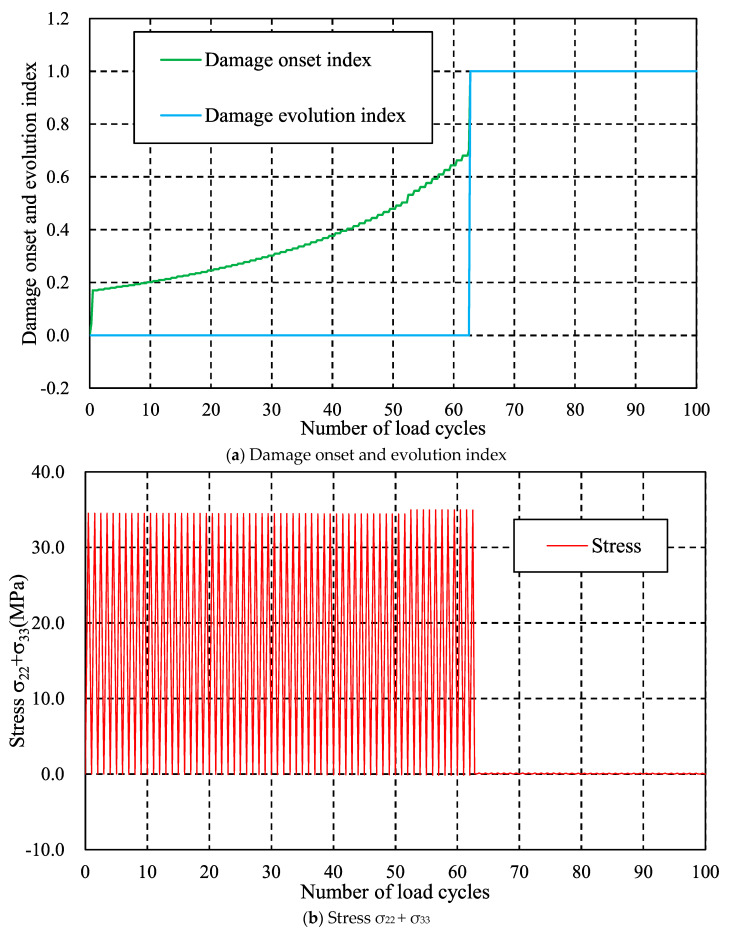
Evolutions of damage onset index, damage evolution index and stress σ_22_ + σ_33_ at first crack when the strain boundary condition is 0.6% and load frequency is fixed at 5 Hz.

**Figure 13 materials-16-00388-f013:**
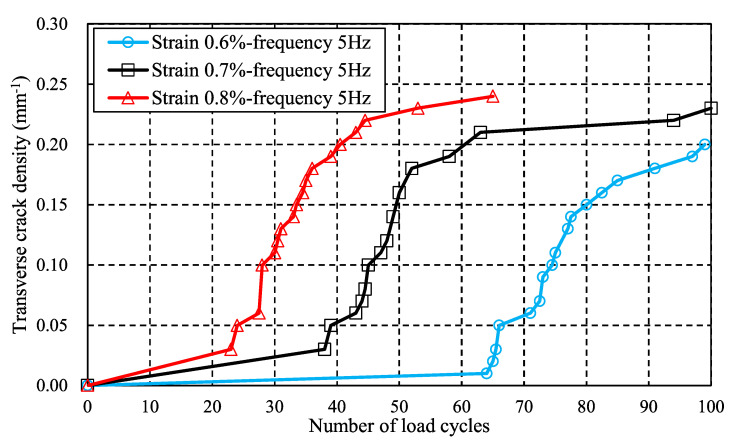
Effect of stress level on transverse crack density: strain boundary condition varies from 0.6% to 0.8% while the load frequency is fixed at 5 Hz.

**Figure 14 materials-16-00388-f014:**
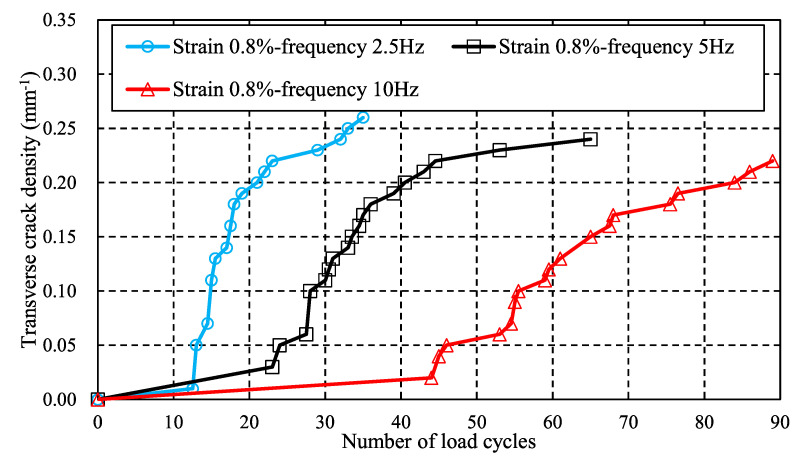
Effect of load frequency on transverse crack density: load frequency varies from 2.5 Hz to 10 Hz while the strain boundary condition is fixed at 0.8%.

**Table 1 materials-16-00388-t001:** Material properties of viscoelastic Maxwell elements [41].

*k*	1	2	3	4	5
$E_{11}^{k}\;$(MPa)	128,000	80	80	80	80
$E_{22}^{k}$, $E_{33\;}^{k}$(MPa)	4290	267	267	267	267
$G_{12}^{k}$, $G_{13}^{k}\;$(MPa)	1810	133	133	133	133
$G_{23}^{k}\;$(MPa)	1610	101	101	101	101
$\eta_{11}^{k}$ (MPa·s)	1 × 10^30^	3.50 × 10^6^	3.00 × 10^6^	3.00 × 10^5^	6.00 × 10^3^
$\eta_{22}^{k}$, $\eta_{33}^{k}\;$(MPa·s)	1 × 10^30^	1.17 × 10^7^	1.00 × 10^7^	1.00 × 10^6^	2.01 × 10^4^
$\eta_{12}^{k}$, $\eta_{13}^{k}\;$(MPa·s)	1 × 10^30^	5.83 × 10^6^	5.00 × 10^6^	5.00 × 10^5^	9.99 × 10^3^
$\eta_{23}^{k}$ (MPa·s)	1 × 10^30^	4.45 × 10^6^	3.81 × 10^6^	3.81 × 10^5^	7.63 × 10^3^

**Table 2 materials-16-00388-t002:** Strength and degradation properties of CFRP laminate [41].

Material Properties	Symbol	Value
Initial axial tensile strength (MPa)	*X* _T,0_	3930
Initial axial compressive strength (MPa)	*X* _C,0_	2775
Initial transverse tensile strength (MPa)	*Y* _T,0_	150
Initial transverse compressive strength (MPa)	*Y* _C,0_	270
Initial axial shear strength (MPa)	*S*_12,0_, *S*_13,0_	117
Initial axial transverse strength (MPa)	*S* _23,0_	117
Initial fiber directional tensile fracture energy (N/mm)	*G* _ft,0_	112.7
Initial fiber directional compressive fracture energy (N/mm)	*G* _fc,0_	25.9
Initial transverse tensile fracture energy (N/mm)	*G* _mt,0_	0.5
Initial transverse compressive fracture energy (N/mm)	*G* _mc,0_	0.5
Degradation coefficient (K·mm^3^/J)	***α*** (*α*_AT_, *α*_AC_, *α*_o_)	300,000

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
