# Peer review of "Numerical and Experimental Studies for Fatigue Damage Accumulation of CFRP Cross-Ply Laminates Based on Entropy Failure Criterion"

_materials, 2022, doi:10.3390/ma16010388_

Round 1

Reviewer 1 Report

This study investigates the transverse cracking behavior of a carbon-fiber-reinforced plastic (CFRP) using an entropy-based failure criterion. The method and findings are worthy publication, however, a number of places shall be improved before accepted for publication.

1.       The title. The authors also do a small fatigue experiment on CFRP laminate, this should be also reflected in the title.

2.       “stress level”. This word is frequently used over the manuscript, however, it is lack of a formal definition. In engineering, we usually call it “stress range”, which is σmax – σmin. Does it have the same definition here?

3.       Introduction. In the first paragraph, the authors mention the need for the fatigue study for FRP from the mechanical eng perspective, which is good. It is better to extend the application to civil eng, where FRPs have also been widely used, and fatigue problem for those structures undergo cyclic loadings. Some examples: Constr Build Mate 2021 295 123544; Frontiers in Materials, 2022, 9, 872055.

4.       Section 3. “Numerical” is an adj. not a noun., authous should change the title to “numerical investigation” or sth similar. In addition, the method in this section is based on the authors previous studies, it shall be further elaborated for the readers without very strong background on this method.

5.       Section 3.3. The authors conducted both experimental and numerical studies in this paper. However, there seems to be a missing link in the numerical results that the authors did not verify the numerical results against experimental results.

6.       Fig. 9 and Fig. 10. Why there are 6 elements along the thickness of the model in Fig. 9, but only 4 elements in Fig. 10?

7.       Section 3.4. The “future plan” may not be of interest of the readers, the authors may rather address the “limitations” of your method.

8.       Some minor editorial comments. The authors should pay attention to the format of paper writting, such as put a space between the text and a bracket.

Author Response

Responses to comments of reviewer 1

The manuscript entitled: Numerical simulation for fatigue damage of CFRP cross-ply laminates based on entropy failure criterion presents interesting research results. In the present study, the authors investigated the transverse cracking behaviour of a carbon fibre reinforced plastic (CFRP) laminate using a fatigue test and an entropy-based failure criterion. The study proposes an entropy-based failure criterion to predict the long-term durability of laminates subjected to cyclic loading. The research results described in the article correspond to the topics of the journal Materials. The following comments will help to improve the manuscript:

  1. 1. In my opinion, the introduction to the topic could be expanded. Manuscripts presenting numerical studies supported by experimental studies of buckling of reinforced polymer composites could be helpful. I suggest reading the following papers presenting stability studies of plate elements made of CFRP.

10.3390/ma15217631

10.3390/ma14112732

10.1016/j.compositesb.2021.109346

Response: The suggested three references are included in the first paragraph of Introduction “In the practical applications, the CFRPs usually suffer from post-buckling under eccentric force [10][11][12] and cyclic loadings.”

  1. 2. What method was used to make the actual composite samples. Please provide the parameters of the manufacturing process.

Response: A new section “2.1 Material manufacturing and damage observation” is added to explain the manufacturing process.

  1. 4. It is recommended that the research position is presented and described. Please insert drawings of the experimental studies.

Response: A new section “2.2 Fatigue loading” is added to describe the experimental studies.

  1. 5. Fig. 4d should be modified, the graduation on the horizontal axis should start at 100 (number of load cycles).

Response: The Fig. 4(d) in the original version is updated based on the comments and re-numbered to Fig. 5(d).

  1. 6. Please specify in which origin or how the mechanical and limiting properties shown in Tables 1 and Tab.2 were determined.

Response: The mechanical and limiting properties shown in Tables 1 and Tab.2 are adopted from the reference [41] and the specification is added at the last sentence in the Section 3.1.

  1. 7. A description of the FEM numerical model should be added:

- how was the composite modelled?

- how were the boundary conditions and loading defined?

- what type of finite element was adopted for the numerical analysis.

Response: The description of the FEM numerical model is discussed in the first paragraph of Section 3.3: “As shown in Fig. 9(a), the dimension of the CFRP structure is 100mm×10mm×6mm and the symmetrical boundary conditions are applied on surfaces ABCD, CDHG and AEHD. In the 90° layer, the transverse tensile strength is assumed to satisfy the cumulative distribution function for the Weibull distribution , where m=20 and s0=90 MPa. A finite element model made of 7777 nodes and 6000 C3D8 elements is shown in Fig. 9(b), in which color denotes material property. In this example, the transverse crack density is determined by dividing the number of transverse cracks by the length (100 mm). To ensure numerical stability, the stress boundary condition is replaced by applying strain boundary conditions, i.e., displacement boundary condition is employed on surface ABFE in Fig. 9(a). The effect of stress levels on results can be reflected by changing the displacement conditions.”

  1. 8. Why were solid elements chosen rather than shell elements dedicated to thin-walled structures?

Response: Comparing with the shell elements, the solid elements simulate more realistic deformation without kinematic assumptions, such as Kirchhoff–Love theory: the normal to the midsurface remains straight and normal.

  1. 9. The font of the legend in the component figures of Figure 9 is not legible, it should be enlarged in Abaqus.

Response: Fig. 9 is updated based on the comments and re-numbered to Fig. 10.

  1. 10. Has the effect of finite element size on the results (mesh test) been investigated in FEM analysis?

Response: The effect of finite element size on the results has not been investigated in the FEM analysis. It is well-known that the fine elements will result in computation burden and some technologies, such as multi-timescale method, will be developed in the next step.

Author Response

Responses to comments of reviewer 2

The fatigue performance of CFRP plate was studied by simulation and experiment. Some research results are important for evaluating the fatigue performance and predicting the fatigue life of CFRP. The authors are advised to further consider the following comments to make the improvement.

  1. 1. The topic does not cover all contents. In addition to numerical simulation, fatigue testing is also a part of this work. It is suggested to further adjust and modify the title.

Response: The title of this paper is modified as “Numerical and Experimental studies for fatigue damage accumulation of CFRP cross-ply laminates based on entropy failure criterion”.

  1. 2. Abstract, please provide some quantitative analysis results on the fatigue performance of CFRP. In addition, the relevant analysis and summary on fatigue damage mechanism and process of CFRP are recommended.

Response: The Abstract is re-written to include the suggestions mentioned above: “The results of fatigue experiments show that the crack accumulation behavior depends on the cyclic number level and frequency, in which two obvious transverse cracks are observed after 104 cyclic loads and 37 transverse cracks occur after 105 cycles.” and “It is discovered that at the edge the stress s22+s33 that is dominant factor for matrix tensile failure mode is greater than interior at the first cycle load, and as stress levels rise, a transverse initial crack forms sooner. However, the initial transverse crack initiation is delayed as load frequencies increase.”

  1. 3. Introduction, the main application fields of carbon fiber reinforced polymer composites have been introduced by the authors in the first paragraph. However, there is no detailed analysis and summary on the composition, performance and advantages of CFRP, such as light weight and high strength, excellent corrosion resistance, fatigue resistance and creep resistance compared with other types of FRP (such as GFRP). The authors can read the following relevant research work to fill this gap. Composite Structures, 2022, 293, 115719. Composite Structures, 2021, 261, 113285.

Response: The first sentence of Introduction is re-written to discuss the advantage of CFRP: “With the development of recent years, composite materials made from two or more constituent materials, such as aramid fiber reinforced plastic (FRP) [1] [2], glass FRP [3], basalt fiber reinforced polymer [4] and carbon fiber reinforced plastics (CFRP) [5][6], have been extensively applied to the automobile, aerospace, civilian and military due to its excellent corrosion resistance, fatigue resistance and creep resistance [7][8].”

  1. 4. Durability and fatigue performance should be further distinguished in the second paragraph of the introduction. Generally, durability refers to environmental effect or service performance and fatigue refers to the effect of cycle loading. There should be some differences between the both. It is suggested that the authors should make adjustments or modifications according to the main research contents of this paper.

Response: The “durability” is replaced by “lifetime” in the second paragraph of the Introduction. In addition, some modifications are made to the distinguish of “durability” and “fatigue performance”.

  1. 5. In addition to fatigue failure and fatigue life in the second paragraph, the fatigue damage mechanism should also be further analyzed and summarized when considering different fatigue factors, such as Polymer testing, 2020; 90: 106761.

Response: The discussion about different fatigue factors is added in the second paragraph of Introduction “Li et al. [28] investigated the effect of various factors, such as elevated temperature and hydraulic pressure, on the property evolution for a carbon/ glass hybrid rod.”.

  1. 6. Please provide a description on the performance parameters, preparation methods and sources of CFRP. Before the fatigue test, the introduction of anchoring method is necessary. Different anchoring methods have different effects on the fatigue properties of composites.

Response: New sections “2.1 Material manufacturing and damage observation” and “2.2 Fatigue loading” are added to describe the preparation methods and fatigue testing process.

  1. 7. Please add some details about fatigue testing, such as type of fatigue testing machine, fatigue testing process, etc.

Response: A new section “2.2 Fatigue loading” is added to describe the fatigue testing machine and fatigue testing process.

  1. 8. In Figure 4 and Figure 5, what methods are used to detect the transverse cracks of CFRP during the fatigue process? How about the reliability and stability of this method?

Response: Figure 4 and Figure 5 are re-numbered to Fig. 5~6. The illustration of the method to detect the transverse cracks is explained at the last sentence of Section 2.1 Material manufacturing and damage observation: “An X-ray machine, M-100S, SOFTEX is used with the applied voltage and current of 14 KVP and 1.5 mA respectively (exposure time is 3 minutes). This damage observation method is a common method used to detect transverse cracks and delamination in the CFRP laminates [45][46][47][48].”

  1. 9. The material parameters in Table 1 and Table 2 are from the reference of 39. Are these parameters consistent with CFRP used in this paper? Please provide relevant explanations.

Response: These parameters are not consistent with CFRP used in this paper. The explanation is added in the last sentence of Section 3.1: “The material parameters are adopted from the reference [41] and listed in the Tab. 1 and 2 since the material properties of the specimen used for fatigue test have not been determined.”

  1. 10. Before analyzing the simulation results, the accuracy of the model should be further verified through the experimental results. Please add relevant comparative analysis between models and experimental results.

Response: There are some limitations of the proposed method. Only 100 cyclic loads are considered owing to the high computational cost of three-dimensional simulation. It is necessary to develop a multi-timescale computational framework to reduce the time cost in the next plan. The heat generation phenomenon under cyclic load is a key factor in estimating the fatigue life of CFRPs, but it is still not implemented into the proposed entropy-based failure criterion. A lot of works needs to be done to improve the accuracy of the model in the next plan.

  1. 11. In Figure 13, please explain why the density of transverse cracks decreases with the increase of fatigue frequency?

Response: Figure 13 is re-numbered to Figure 14. Until now, the phenomenon that the density of transverse cracks decreases with the increase of fatigue frequency has not been explained theoretically.

  1. 12. Some quantitative results are suggested to add into the conclusions.

Response: The conclusions are re-written to include quantitative results: “Two obvious transverse cracks are observed after 104 cyclic loads and 37 transverse cracks occur after 105 cycles in the experimental test. The final numbers of transverse cracks decrease from 29 to 11 when the load frequency increases from 5 Hz to 10 Hz.” and “When the load frequency is fixed as 5 Hz, the initial transverse cracks form after 63 cyclic loads when the strain boundary condition is 0.6%, and those of 0.7% and 0.8% are 38 and 23. In addition, as load frequency increase from 2.5 Hz to 10 Hz, the numbers of cyclic loads where the initial crack forms increase from 13 to 44.”

Reviewer 3 Report

The fatigue performance of CFRP plate was studied by simulation and experiment. Some research results are important for evaluating the fatigue performance and predicting the fatigue life of CFRP. The authors are advised to further consider the following comments to make the improvement. 

1.     The topic does not cover all contents. In addition to numerical simulation, fatigue testing is also a part of this work. It is suggested to further adjust and modify the title.

2.     Abstract, please provide some quantitative analysis results on the fatigue performance of CFRP. In addition, the relevant analysis and summary on fatigue damage mechanism and process of CFRP are recommended.

3.     Introduction, the main application fields of carbon fiber reinforced polymer composites have been introduced by the authors in the first paragraph. However, there is no detailed analysis and summary on the composition, performance and advantages of CFRP, such as light weight and high strength, excellent corrosion resistance, fatigue resistance and creep resistance compared with other types of FRP (such as GFRP). The authors can read the following relevant research work to fill this gap. Composite Structures, 2022, 293, 115719. Composite Structures, 2021, 261, 113285.

4.     Durability and fatigue performance should be further distinguished in the second paragraph of the introduction. Generally, durability refers to environmental effect or service performance and fatigue refers to the effect of cycle loading. There should be some differences between the both. It is suggested that the authors should make adjustments or modifications according to the main research contents of this paper.

5.     In addition to fatigue failure and fatigue life in the second paragraph, the fatigue damage mechanism should also be further analyzed and summarized when considering different fatigue factors, such as Polymer testing, 2020; 90: 106761.

6.     Please provide a description on the performance parameters, preparation methods and sources of CFRP. Before the fatigue test, the introduction of anchoring method is necessary. Different anchoring methods have different effects on the fatigue properties of composites.

7.     Please add some details about fatigue testing, such as type of fatigue testing machine, fatigue testing process, etc.

8.     In Figure 4 and Figure 5, what methods are used to detect the transverse cracks of CFRP during the fatigue process? How about the reliability and stability of this method?

9.     The material parameters in Table 1 and Table 2 are from the reference of 39. Are these parameters consistent with CFRP used in this paper? Please provide relevant explanations.

10.  Before analyzing the simulation results, the accuracy of the model should be further verified through the experimental results. Please add relevant comparative analysis between models and experimental results.

11.  In Figure 13, please explain why the density of transverse cracks decreases with the increase of fatigue frequency?

12.  Some quantitative results are suggested to add into the conclusions.

Author Response

Responses to comments of reviewer 3

General

The publication describes the investigation of damage mechanisms under cyclic loading of a cross-ply laminate. In addition to the experimental investigations, FEM analyses are carried out to describe the failure mechanism. The entropy failure criterion is used within the simulations. Experimental work and simulation showed that the frequency and load level significantly influence the damage propagation, which the classical Paris' law cannot address.

Language

In terms of language, this article is written in a readable and understandable style.

Keywords

In general, the keywords match the content of the publication. In most cases, the search for relevant publications uses catchwords that refer to the title and the keywords. To increase the search results or the findability, it is helpful if the keywords are different from the title of the publication. However, this is only a note. The keywords used in the publication match the content well.

Abstract

The abstract provides a good overview of the content of the publication. I do not see any necessary changes.

Chapters one to four

  1. 1. In general, the authors use a lot of references in the first chapter of their publication. In the first section, general statements are supported by six references each. Here, the decisive reference, or two decisive references, should be identified and indicated.

Response: The decisive references are improved and two or three references are cited in each topic.

  1. 2. There is a linguistic error on page 2 in the first paragraph. The authors write "CFRP play".

Response: The linguistic error on page 2 is corrected to “ply”.

  1. 3. In chapter two, the authors describe the tension-tension experiments and state that the applied stresses are between 20 and 200 MPa. Here it would be essential to know the ultimate tensile strength of the tested material.

Response: A new section 2.2 is added to explain the strength parameters of the specimen and the laminate’s average maximum tensile strength is 647 MPa where the transverse cracks initiated from about the stress level of 250 MPa.

  1. 4. The same issue can be found, for example, in the caption of figure 4. For the reader, the information about the stress level is important; for example, "Transverse crack distribution under various cyclic load numbers at a maximum stress of 40% of the tensile strength".

Response: Figure 4 is titled as “Transverse crack distribution under various cyclic load number at a maximum stress of 30% of the tensile strength” and re-numbered to Fig. 5.

  1. 5. In the figures showing the transverse cracks, the authors should insert scales to make it easier to overview the space between the cracks.

Response: The figures showing the transverse cracks are updated.

  1. 6. The authors should briefly explain in one or two sentences on page 8 the strength degradation parameter and how they determined the value used.

Response: The strength degradation parameters are explained in the last paragraph of Section 3.2: “The strength degradation parameters a(aAT, aAT and aO) [8] [41] are taken as a slightly larger constant 300000K·mm3/J, which will be further studied in next plan.”

  1. 7. The results of the simulations qualitatively agree with those of the experiments and show the same trend. However, the quantitative results show a significant discrepancy with the simulative results. Could the authors provide further information on why this discrepancy occurs?

Response: There are some limitations in the current entropy-based failure criterion. On the one hand, the heat generation phenomenon under cyclic load is a key factor in estimating the fatigue life of CFRPs, but it is not considered. On the other hand, the stiffness degradation caused by the entropy generation is not implemented into UMAT. These are discussed in Section 3.4.

Reviewer 4 Report

Dear authors,

attached is a PDF File with my comments.

Author Response

Responses to comments of reviewer 4

The article covers the topic of Numerical simulation for fatigue damage of CFRP cross-ply laminates based on entropy failure criterion. The subject and the supporting experiments are informative but the cohesion and structure of manuscript must be improved. In my opinion, article presents not high novelty and the quality of paper is relatively poor. I suggest that article should be improved before potential publication. Suggestions as follows:

  1. 1. In abstract part please stronger underline major results of this research (little more precisely - showing quantitative results, with details).

Response: The Abstract is re-written and some quantitative results are included: “The results of fatigue experiments show that the crack accumulation behavior depends on the cyclic number level and frequency, in which two obvious transverse cracks are observed after 104 cyclic loads and 37 transverse cracks occur after 105 cycles. The final numbers of transverse cracks decrease from 29 to 11 when the load frequency increases from 5 Hz to 10 Hz.”

  1. 2. In introduction part please show literature survey related to the topic. It is recommended to show results (numerical and laboratory) from last years related the topic of determination of mechanic paramenters of different composites (mainly CFRP, but also AFRP, GFRP, BFRP etc.)

Response: The topic of determination of mechanic parameters about AFRP, GFRP, and BFRP is included in the first paragraph of Introduction: “With the development of recent years, composite materials made from two or more constituent materials, such as aramid fiber reinforced plastic (FRP) [1][2], glass FRP [3], basalt fiber reinforced polymer [4] and carbon fiber reinforced plastics (CFRP) [5][6][7], have been extensively applied to the automobile, aerospace, civilian and military due to its excellent corrosion resistance, fatigue resistance and creep resistance [8][9][10][11].”

  1. 3. I suggest to add separated point - Research significance - Please describe here the main essence of the research. What was the inspiration for such an analysis? Why presented studies are so important?

Response: The fourth and fifth paragraph in the Introduction are used to explain the advantage of the entropy-based failure criterion and inspiration of this study: “Although the entropy-based failure criterion has been widely utilized to successfully estimate the fatigue life of metal components, its application to investigate the long-term lifetime of CFRP under cyclic loading is limited.”

  1. 4. The key problem is that the article in its current version is not similar to a good research paper, at best to a research note. One of major drawbacks are as follows: lack of in depth literature survey and research significance, construction of paper (lack of materials and methods point), superficial analysis of the results and poor conclusions.

Response: The literature review is updated and research significance is explained in the fourth and fifth paragraphs in the Introduction. In the numerical results, the effects of stress level and load frequency on the transverse crack accumulation behaviors are studied. The proposed method’s results agree well with those of the existing experimental method qualitatively. In addition, the proposed entropy-based failure criterion can account for the effect of load frequency on transverse crack growth rate, which cannot be addressed by the well-known Paris’ law. These may be significant contributions to this study.

  1. 5. Point "Tension-tension fatigue experiment of CFRP" should be renamed and construction should be improved. This is not acceptable to show many figures without any text between them and without detailed explanation of these elements (for instance Figures 1-5!). Materials and methods should be shown precisely.

Response: New sections “2.1 Material manufacturing and damage observation” and “2.2 Fatigue loading” are added to describe the preparation methods and fatigue testing process.

  1. 6. Quality of figure 10 must be improved, legend is not visible.

Response: Figure 10 is updated to make the legend visible and re-numbered to Fig. 11.

  1. 7. Numerical results should be better analyzed.

Response: The discussions of the numerical results are further improved to demonstrate the novelty and merits of this study: “It is worth noting that, although some empirical formulations, such as Paris’ law [27] based on the energy release rate, are proposed to predict the crack accumulation phenomenon, the effect of load frequency on transverse cracking behavior cannot be addressed. However, the effect of load frequency can be addressed by the proposed entropy-based failure criterion. Thus, the proposed that can account for the effect of load frequency on transverse crack growth behaviors of CFRPs than the conventional methods.”

  1. 8. How did you deteremine mesh in models?

Response: The FEM model is explained in the first paragraph of Section 3.3: “A finite element model made of 7777 nodes and 6000 C3D8 elements is shown in Fig. 9(b), in which color denotes material property and the size of element is 1mm×1mm×1mm.”

  1. 9. The conclusion part must be more accurate (and presented by points).

Response: The conclusions are re-written by points to make the conclusions accurate.

  1. 10. It is recommended to indicate potential application of research results in civil engineering.

Response: The potential application of research results in civil engineering is indicated in the Section 3.4: “Although the proposed method is only applied to the transverse cracking behavior of cross-ply laminates in this study, the quasi-isotropic laminates [56][57][58][59][60], delamination caused by the transverse cracks, and components in the civil engineering [61] can also be analyzed.”

Reviewer 5 Report

The article covers the topic of Numerical simulation for fatigue damage of CFRP cross-ply laminates based on entropy failure criterion.
The subject and the supporting experiments are informative but the cohesion and structure of manuscript must be improved.
In my opinion, article presents not high novelty and the quality of paper is relatively poor.

I suggest that article should be improved before potential publication.
Suggestions as follows:
1. In abstract part please stronger underline major results of this research (little more precisely - showing quantitative results, with details).
2. In introduction part please show literature survey related to the topic. It is recommended to show results (numerical and laboratory) from last years relatedthe topic of determination of mechanic paramenters of different composites (mainly CFRP, but also AFRP, GFRP, BFRP etc.)
3. I suggest to add separated point - Research significance - Please describe here the main essence of the research. What was the inspiration for such an analysis? Why presented studies are so important?
4. The key problem is that the article in its current version is not similar to a good research paper, at best to a research note.
One of major drawbacks are as follows: lack of in depth literature survey and research significance, construction of paper (lack of
materials and methods point), superficial analysis of the results and poor conclusions.
5. Point "Tension-tension fatigue experiment of CFRP" should be renamed and construction should be improved.
This is not acceptable to show many figures without any text between them and without detailed explanation of these elements (for instance Figures 1-5!).
Materials and methods should be shown precisely.
6. Quality of figure 10 must be improved, legend is not visible.
7. Numerical results should be better analyzed.
8. How did you deteremine mesh in models?
9. The conclusion part must be more accurate (and presented by points).   
10. It is recommended to indicate potential application of research results in civil engineering.

Author Response

(The authors gave the same response as above.)

Round 2

Reviewer 2 Report

Accept in present form.

Reviewer 3 Report

It is suggested to accept it. 

Reviewer 5 Report

The majority of remarks have been considered by authors. Errors have been eliminated.
The authors responded to comments of the reviewer thoroughly.
The current version is more satisfactory for reviewer.